# Potential Factors Associated with Healthcare Utilization for Balance Problems in Community-Dwelling Adults within the United States: A Narrative Review

**DOI:** 10.3390/healthcare11172398

**Published:** 2023-08-26

**Authors:** Shweta Kapur, Kwame S. Sakyi, Prateek Lohia, Daniel J. Goble

**Affiliations:** 1School of Health Sciences, Oakland University, Rochester, MI 48309, USA; skapur2@oakland.edu (S.K.); ksakyi@oakland.edu (K.S.S.); dgoble@oakland.edu (D.J.G.); 2Department of Internal Medicine, Wayne State University, Detroit, MI 48201, USA

**Keywords:** balance, healthcare utilization, falls, barriers, community dwelling adults, fall prevention

## Abstract

Falls are the leading cause of mortality and chronic disability in elderly adults. There are effective fall prevention interventions available. But only a fraction of the individuals with balance/dizziness problems are seeking timely help from the healthcare system. Current literature confirms the underutilization of healthcare services for the management of balance problems in adults, especially older adults. This review article explores factors associated with healthcare utilization as guided by the Andersen Healthcare Utilization Model, a framework frequently used to explore the factors leading to the use of health services. Age, sex, race/ethnicity, BMI, and comorbidities have been identified as some of the potential predisposing factors; socioeconomic status, health insurance, and access to primary care are the enabling and disabling factors; and severity of balance problem, perceived illness, and its impact on daily activities are the factors affecting need for care associated with healthcare utilization for balance or dizziness problems. Knowledge about these barriers can help direct efforts towards improved screening of vulnerable individuals, better access to care, and education regarding effective fall prevention interventions for those who are at risk for underutilization. This can aid in timely identification and management of balance problems, thereby reducing the incidence of falls.

## 1. Introduction

### 1.1. Background

Balance, the ability to maintain the body’s center of mass over its base of support, is maintained by a complex set of interactions between the sensory and motor systems [1]. Injury, disease, or aging can affect these interactions and result in balance disorders [2]. Balance disorders, which are highly prevalent in the elderly [3,4], can make the person unsteady, off-balance, or dizzy, and may result in a fall. In the United States (US), falls are a leading cause of mortality and chronic disability in community-dwelling adults, especially those over 65 years of age [5].

The Center for Disease Control and Prevention defines falls as unexpected or unintentional dropping to a lower surface (i.e., the floor), from a standing, seating, walking, or bending position [6]. A study by Penger et al. reported dizziness, which is one of the risk factors for falls, to be a common complaint among adults more than 50 years old [7,8]. Since dizziness is responsive to rehabilitation, this study also called upon public health officials to raise awareness of dizziness in this population [8]. The World Health Organization has reported that falls are the second leading cause of unintentional injury deaths worldwide [9]. Approximately 37.3 million falls occurring worldwide require medical attention each year [9]. In a study of 6785 patients by Agrawal et al., about 35% of Americans over the age of 40 years had objective evidence of vestibular dysfunction [10]. In the same study, about 32% of the individuals who reported having no symptoms of dizziness, were found to have subclinical vestibular dysfunction which correlated with an increased number of falls [10]. Screening of such individuals and early treatment may be an avenue to prevent falls.

### 1.2. Burden on Healthcare

Falls are the number one cause of nonfatal unintentional emergency department visits in the US, according to the National Center for Injury Prevention and Control [11]. A recent cross-sectional study of fall injuries by Hoffman et al. using the data from years 2016 to 2019, reported an increase of 1.5% per year in fall-related injuries [3]. It has been estimated that there were $616.5 million in direct medical costs for fatal fall-related injuries and $30.3 billion for non-fatal injuries in 2012 [12]. Given the 12-fold increase in the likelihood of falls in symptomatic patients with vestibular dysfunction, Agrawal et al. emphasized the need for screening, diagnosing, and treating patients with vestibular deficits to reduce the burden on healthcare due to fall-related injuries [10]. Early management of balance problems, both vestibular and non-vestibular, that are responsive to treatment can help reduce this ever-increasing burden of falls on healthcare.

### 1.3. Healthcare Utilization

According to the United States Department of Health and Human Services, around 70% of adults with moderate to severe balance or dizziness problems saw a healthcare specialist in 2016 [13]. Since the individuals with mild balance or dizziness problems might not have pronounced clinical symptoms or might even be unaware of the problem [10], it can be postulated that the number of individuals with mild balance or dizziness problems seeking healthcare services would be even lower. Lin et al. conducted a cross-sectional analysis and reported that one in every five elderly individuals experiences dizziness or balance problems every year, but only about 50% of these individuals sought help from the healthcare system [4]. Although there is a plethora of studies discussing falls and their burden on healthcare and the efficacy of fall prevention programs [3,11,12], there is limited data on the factors associated with the under-utilization of healthcare services in individuals with balance problems.

Only one known study has attempted to identify the factors associated with inpatient healthcare utilization resulting from falls in community-dwelling elderly individuals within the US. In that study, Choi et al. found that certain injury characteristics, location of falls (home vs. outside), and factors such as living alone were more likely to be associated with hospitalization following fall injuries [14]. This study also reported that almost three-quarters of the individuals experiencing falls sought treatment in outpatient settings [14]. This study only looked, however, at the healthcare utilization in individuals who had already sustained fall-related injuries. It remains unclear to what extent including all individuals who report balance or dizziness problems would affect the analysis. Further, barriers preventing these individuals from seeking help before the condition results in a serious injury are yet unknown.

### 1.4. Objective and Significance

The objective of this article is to review the existing literature to explore the potential factors that might be associated with the utilization of healthcare services for balance problems in community-dwelling adults in the US. This information can help facilitate future research in this domain investigating the healthcare utilization for balance problems. Given the scarcity of existing studies directly discussing the healthcare utilization for balance problems before a serious balance event or fall occurs, a list of potential factors can be useful for further studies aiming to investigate these associations with the utilization of healthcare services. The knowledge about the barriers to healthcare utilization for balance problems in community-dwelling adults can form the basis of future healthcare interventions to improve access for these patients. This approach has the potential to reduce falls when the responsible conditions are identified and treated before they result in a loss of balance or a serious fall. In addition, these findings might help in the more efficient allocation of healthcare resources. It can help in focusing the efforts on effective screening of the vulnerable individuals, on improving the access to care and on educating, those who are at higher risk for underutilization, about the effective fall prevention interventions available.

## 2. Framework

Different frameworks have been utilized in the past to analyze and understand factors associated with healthcare resource utilization. The Andersen Healthcare Utilization Model is one of the most frequently used frameworks [15], which can be used to explore and demonstrate the factors leading to the use of health services [16].

The Andersen Healthcare Utilization Model has evolved since its conception in the 1960s, but the primary focus of this model remains unchanged, involving three categories of the factors recognized as determinants of health service utilization. These categories include predisposing factors, enabling and disabling factors, and factors affecting the need for care [16,17].

## 3. Predisposing Factors

Predisposing factors are the factors that exist before the onset of illness and predispose an individual to be more or less likely to seek healthcare services [17]. They include demographic factors (such as age, sex, marital status, past illness/comorbidities), social factors (education, race, occupation), and beliefs (values concerning health, attitude towards health services, etc.) [15,16,17].

### 3.1. Age

Falls are more common in adults over the age of 65 years [12,18]. Older adults are also responsible for higher utilization of healthcare services [19]. A study done by the Centers for Disease Control and Prevention (CDC), USA recorded 24,190 fatal falls and 3.2 million medically treated non-fatal falls in 2012 using the data from national databases [12]. In this observational study, Burns et al. reported a higher incidence of falls and an increase in the direct medical costs related to fatal and non-fatal fall injuries with increasing age [12]. Another retrospective study looking at the utilization of ambulatory medical care services by elderly individuals was done on 123,224 patients by Bussche et al. [20] This study reported similar findings of high utilization of healthcare in the elderly individuals, and among the elderly, those over the age of 75 years had higher utilization of healthcare compared to those between 65 to 75 years of age [20]. So, older age is associated with a higher incidence of falls as well as a higher use of healthcare services.

### 3.2. Sex

Studies from different parts of the world have explored the association between sex and the incidence of falls, and the association between sex and healthcare utilization. Women have greater loss of bone mineral density with age compared to men which increases their likelihood of experiencing fall related injuries [21]. The incidence of falls and related costs have been reported by the CDC, USA, to be higher in women compared to men [12]. The incidence of fall-related injuries has been reported to be almost twice as high in women than in men [12,18]. These findings have been confirmed in a retrospective study by Peel et al. in Australia on 2090 patients over the age of 65 years, hospitalized with fall-related injuries over the study period of 12 months [18]. One limitation of population-based studies can be the generalization of the results to a different geographical location, given the differences in the characteristics of the study participants; so, looking at studies conducted in different parts of the globe can give an idea about the trends worldwide. Higher healthcare expenditures have been reported in women compared to men [22]. A study using historical cohort study design on cross-sectional data reported that women are less likely to receive preventative health screening and regular care [19]. These studies used different clinical populations but point to a similar trend that women have a higher incidence of falls and related injuries but are less likely to receive preventative health screening.

### 3.3. Race/Ethnicity

Racial and ethnic inequalities have been frequently highlighted in healthcare utilization. Decreased use of outpatient medical services and increased use of the emergency room and inpatient services by Blacks and Hispanics compared to Whites and non-Hispanics respectively have been well documented in the literature [23,24]. Dark et al. conducted a retrospective study and found that the odds of non-Hispanic Black respondents reporting to the emergency department and/or needing inpatient utilization were 2.39 times the odds of non-Hispanic White respondents for the same [23]. However, this study looked at the patients with comorbid anxiety disorder and cardiometabolic syndrome. Another observational study that specifically looked at the fall related hip fractures, reported that non-Hispanic Blacks were more likely to experience longer inpatient stays and delays in care compared to non-Hispanic Whites [25]. In another retrospective study in the US analyzing nationally representative data of patients with neurological conditions such as Parkinson’s disease, multiple sclerosis, headache, cerebrovascular disease, and epilepsy, Saadi et al. reported that Blacks were 30% less likely to see an outpatient neurologist compared to Whites [24]. The same study also reported that Hispanics were 40% less likely to seek regular care compared to the non-Hispanic population [24]. Hence it can be postulated that the Black and Hispanic population might be less likely to use healthcare services for balance and dizziness problems, until the latter results in a loss of balance event or a serious fall warranting the need for emergency or inpatient services.

### 3.4. Body Mass Index (BMI)

A higher body mass index has been associated with both the increased risk of fall-related injuries and the higher utilization of healthcare services in multiple separate studies [26,27,28]. Some studies have reported higher risk of falling in individuals with both low and high BMI compared to those with normal BMI [29]. A study by Finkelstein et al. reported a higher probability of sustaining a fall-related injury in individuals with high BMI [27]. This study was a cross-sectional analysis done on the data for 42,304 adults representing the non-institutionalized population of the US [27]. Individuals with a BMI of 40 kg/m^2^ or higher had 1.79 times higher odds of sustaining a fall-related injury compared to individuals with a normal BMI [27]. One limitation of this study is that it relied on self-reported height and weight for the BMI calculation. However, high BMI is also associated with multimorbidity and a higher incidence of diabetes, which can in turn make these individuals more prone to falls and related injuries [28]. Edwards et al. used the data for 33,882 adults in Norway to look at both the annual utilization and the lifetime utilization of healthcare services [26]. This study reported that the number of healthcare contacts increased from 3.1/year in individuals with normal BMI to 6.1/year in individuals with class 2 and 3 obesity [26]. Even after considering the early mortality associated with high BMI, lifetime utilization was found to be higher in obese individuals [26]. However, this study explored the overall healthcare utilization and not the healthcare utilization due to fall-related injuries per se.

### 3.5. Comorbidities

The presence of multiple comorbidities has been associated with higher utilization of healthcare services [30]. Certain comorbidities also increase the risk of falls. A meta-analysis of studies from Australia, Sweden, the United Kingdom, and the US looked at 537 participants with 1721 reported falls and found a high risk of falls in patients with Multiple Sclerosis [31]. Another review on balance and falls in Multiple Sclerosis patients by Cameron et al. in the USA reported that 50–80% of these patients have balance and gait dysfunction and more than 50% fall at least once every year [32]. Individuals with neurological disorders have been reported to have more frequent and more severe falls [33]. Recurrent and disabling falls have been commonly observed in patients with Parkinson’s disease [34,35]. A study conducted by Rudzinska et al. on Parkinson’s disease patients reported that these patients fell three times more frequently compared to age and sex-matched controls [35]. Homann et al. found that patients with hyperkinetic movement disorders including Huntington’s disease, restless leg syndrome, and spinocerebellar ataxias had increased fall risk [33]. According to a meta-analysis of 74 studies, individuals with gait problems or those using walking aids were 2 to 3 times more likely to experience a fall [36]. Patients with Diabetes Mellitus (DM) have been reported to have higher fall risk; the increased risk of falling due to DM was much more pronounced in insulin-treated patients compared to non- insulin-treated patients [28]. Differences in the utilization of healthcare services have been reported by a population-based study on 21,277 diabetic patients [37]. Patients with DM had higher utilization of services in women compared to men [37]. Hence, the presence of certain comorbidities can be associated with both a higher incidence of falls and a higher utilization of healthcare.

## 4. Enabling and Disabling Factors

Factors that must be available for an individual to use healthcare services have been referred to as enabling factors, and the factors that prevent an individual from using such services are disabling factors. Enabling and disabling factors include socio-economic status, place of residence, health insurance, and access to primary care health services [15,16,17].

### 4.1. Socioeconomic Status

Differences in socio-economic status can result in differences in the ability to cover treatment costs and co-pays, along with the ability to take time off from work for proper evaluation and treatment. Individuals with low socioeconomic status have also described factors such as cost, transportation, poor health education, and lack of a social network as barriers to optimal healthcare utilization [38]. Several studies have reported a socioeconomic gradient in the utilization of healthcare services [39,40,41]. Hence individuals from low socioeconomic classes might be at a greater risk of underutilization of healthcare services for balance problems unless the problem results in a serious event. After the balance problem results in a serious event or disabling fall, these individuals might have a higher healthcare utilization as higher utilization of healthcare services has been reported after surgical procedures in socioeconomically disadvantaged groups [40].

### 4.2. Health Insurance

Healthcare utilization may be determined by the availability of health insurance and the type of insurance the individual has [41,42]. In different clinical populations, utilization of healthcare services has been found to be higher in individuals with public insurance compared to those with private insurance [41,43]. Singh et al. reported that the individuals with public insurance had 1.40 to 1.77 times higher odds of discharge to an inpatient facility instead of home, post hospitalization compared to those with private insurance [41]. A recent study exploring healthcare spending in the US for falls reported that public insurances financed most of the healthcare spending in the elderly individuals over the age of 65 years, whereas in younger individuals most of the spending involved the individuals with private insurances [44]. So, while it seems reasonable to assume that individuals without insurance may be at a higher risk of underutilization of healthcare services for balance problems, more studies are needed to ascertain if the type of insurance has any association with healthcare utilization for balance problems.

### 4.3. Access to Primary Care Services

Timely access to primary care services may assist in the timely diagnosis of the balance problem and efficient management of the problem before it results in a fall. Significant potential for the reduced long-term burden on healthcare and reduced costs have been reported with increased access to primary care services [45,46]. However, a secondary analysis of data from 11 countries reported that about one in five adults experience barriers in accessing primary care services and females were more likely to experience this [45]. Individuals residing in medically underserved areas with limited access to primary care services have been known to have worse health outcomes once they develop a condition that warrants hospitalization [47]. One might expect that those individuals who have not sought any primary care services in the past year could be less likely to seek help for balance problems.

## 5. Factors Affecting Need for Care

Factors associated with the need for care include the level or severity of illness, perceived illness, and its impact on daily activities [15,16,17]. These are the factors that represent the actual and perceived need for care.

### 5.1. Severity of Balance Problem

Healthcare utilization is expected to increase with the increase in the severity of the illness experienced by the individual [48,49]. However, a decrease in the use of recommended care with increasing levels of activity limitation has also been reported [50]. So, it is debatable if a more pronounced balance problem would result in an increased or decreased utilization of healthcare services.

### 5.2. Perceived Illness

Perception of the physical problem can vary with the individual. Secondary analysis of the data from a controlled trial reported that greater patient perception of fall severity in the elderly has been associated with better quality of care the patient received [51]. However, there is a scarcity of literature on the association between patients’ perception of their balance problem and the utilization of healthcare services for the same. Patients perceiving their balance problem to be severe might be expected to seek help from the healthcare system compared to those who perceive their problem to be trivial or small.

### 5.3. Impact on Daily Activities

The impact of balance problems and the resulting falls can vary greatly among individuals. It can range from mild restrictions on the personal freedom to total institutionalization [52]. In a population based cross-sectional study, Taburee et al. reported falls to be one of the predictors of health-related quality of life in community-dwelling adults [53]. Recurrent fallers have also been reported to have poor physical performance and a poor quality of life compared to single fallers [54]. A more pronounced impact of the balance problems on daily activities might warrant an individual to seek help for the management of the problem.

## 6. Conclusions

To summarize, the current literature supports the existence of underutilization of healthcare services for management of balance problems in adults, especially older adults, with balance or dizziness problems. Certain factors may be preventing these individuals from using healthcare services, and identification of such factors would aid in increasing healthcare access and reducing the incidence of falls and related injuries. There is scarcity of literature reporting factors associated with healthcare utilization for balance problems, especially before these problems result in a serious event or fall. The potential factors identified upon the review of literature that may be associated with healthcare utilization for balance problems in community-dwelling adults have been summarized in the Figure 1. Age, sex, race/ethnicity, BMI, and comorbidities have been identified as some of the potential predisposing factors; socioeconomic status, health insurance, and access to primary care are the enabling and disabling factors; and severity of balance problem, perceived illness, and its impact on daily activities are the factors affecting need for care that may be associated with healthcare utilization for balance or dizziness problems.

## 7. Future Directions

Further research using population-based models to confirm these associations is warranted to identify and tackle the barriers to healthcare utilization for balance problems, so that burden of fall related injuries can be reduced with timely intervention and management.

## Figures and Tables

**Figure 1 healthcare-11-02398-f001:**
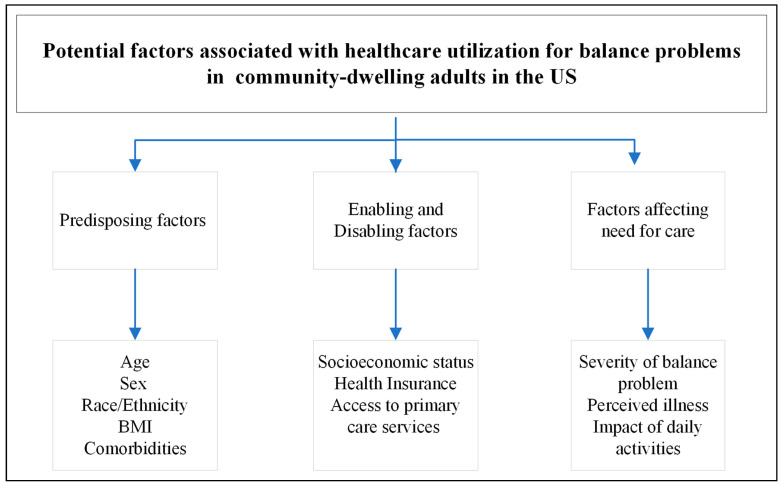
Potential factors identified upon literature review associated with healthcare utilization for balance problems in community-dwelling adults.

## Data Availability

Not applicable.

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
