# Peer review of "Potential Factors Associated with Healthcare Utilization for Balance Problems in Community-Dwelling Adults within the United States: A Narrative Review"

_healthcare, 2023, doi:10.3390/healthcare11172398_

Round 1
Reviewer 1 Report
The manuscript entitled "Potential Factors Associated with Healthcare Utilization for Balance Problems in Community-dwelling Adults within the United States: A Traditional Review" examines the factors contributing to healthcare utilization for balance problems in older adults. The study investigates various factors, including age, gender, BMI, socioeconomic status, and disease severity. The authors conclude that proactive identification of these risk factors can effectively prevent falls and injuries in older individuals. However, the manuscript primarily provides descriptive information without critical appraisal, limiting its contribution to the existing body of evidence. The reviewer suggests the following points to improve the manuscript:
1. The authors did not present concrete figures to evaluate the risk factors. It is crucial to include results from statistical analyses, such as odds ratios, from previous studies that assessed each factor. Without these figures, readers may not grasp the strength of the association between each factor and healthcare utilization for balance problems.
2. The authors solely focus on the healthcare system within the United States. It is recommended to broaden the discussion by comparing healthcare systems across different countries and proposing future directions in the context of aging societies. This comparative approach would provide valuable insights beyond the scope of a single country.
Reviewer 2 Report
line 15: It is Andersen, from Ronald M. Andersen, and not Anderson.
(lines 66-67): Are there evidence based on the literature supporting such postulation? If so, should cited; if not, Authors should present a rationale that supports the statement.
(lines 135-137): The finding referred should be linked to gender differences through the aging process, which lead women to a greater bone mineral density loss, comparatively to men, resulting in a higher likehood to experience hip, vertebral or forearm fracture, consenquently implying higher treatment costs.
Reading suggestion:
Daly RM, Rosengren BE, Alwis G, Ahlborg HG, Sernbo I, Karlsson MK. Gender specific age-related changes in bone density, muscle strength and functional performance in the elderly: a-10 year prospective population-based study. BMC Geriatr. 2013 Jul 6;13:71. doi: 10.1186/1471-2318-13-71. PMID: 23829776; PMCID: PMC3716823.
Colón-Emeric CS, Saag KG. Osteoporotic fractures in older adults. Best Pract Res Clin Rheumatol. 2006 Aug;20(4):695-706. doi: 10.1016/j.berh.2006.04.004. PMID: 16979533; PMCID: PMC1839833.
(lines 153-157): Although it refers to the using medical services issue, the reasons that led these patients to resort to such services are somewhat outside the scope of the study, which focuses on causes related to falls. In my opinion, it should not be quoted.
(lines 167-174): None of the 3 studies referred objectively mention falls by obese people as factor for seeking health care services or even to the associated costs to its occurence, which end up to disperse reader´s attention from the main goal of the study. I guess it should be addressed the tendency obese people experience concerning to falls.
Reading suggestion:
Hooker ER, Shrestha S, Lee CG, Cawthon PM, Abrahamson M, Ensrud K, Stefanick ML, Dam TT, Marshall LM, Orwoll ES, Nielson CM; Osteoporotic Fractures in Men (MrOS) Study. Obesity and Falls in a Prospective Study of Older Men: The Osteoporotic Fractures in Men Study. J Aging Health. 2017 Oct;29(7):1235-1250. doi: 10.1177/0898264316660412. Epub 2016 Jul 27. PMID: 27469600; PMCID: PMC5773405.
Caterina Trevisan, Alessio Crippa, Stina Ek, Anna-Karin Welmer, Giuseppe Sergi, Stefania Maggi, Enzo Manzato, Jennifer W. Bea, Jane A. Cauley, Evelyne Decullier, Vasant Hirani, Michael J. LaMonte, Cora E. Lewis, Anne-Marie Schott, Nicola Orsini, Debora Rizzuto,
Nutritional Status, Body Mass Index, and the Risk of Falls in Community-Dwelling Older Adults: A Systematic Review and Meta-Analysis, Journal of the American Medical Directors Association,
Volume 20, Issue 5, 2019, Pages 569-582.e7, ISSN 1525-8610,
https://doi.org/10.1016/j.jamda.2018.10.027.
(https://www.sciencedirect.com/science/article/pii/S1525861018306078)
Table: It should be mentioned below the table as footnote its intelectual property, which may be a manuscript, book, website, etc., from it was retrieved or the Authors themselves.
Figure 1: unless it is a journal guidance, the title should be placed above the table and beyond that it is not according to APA 7th edition.
Round 2
Reviewer 1 Report
The authors have appropriately addressed my comments.
The reviewer has no additional comments to revise in the manuscript.
Author Response
We appreciate the time and effort that you have dedicated to providing your valuable feedback on our manuscript. We are glad to note that we have appropriately addressed all your comments.
Reviewer 2 Report
Reading the revised manuscript, It seems clear the Authors' availability to accept and incorporate both the comments and the reading suggestions presented by the reviewer, thus contributing to a globally improvement of the text. The fluidity and objectivity evident throughout the different manuscript sections became its approach coeherent and, above all, aligned with the scientific assumptions of a research report, that, in my opinion, ensure the necessary requirements to be published in Healthcare journal.
Author Response
We are glad to note that we have appropriately addressed all your comments. We appreciate the time and effort that you have dedicated to providing your valuable feedback on our manuscript and helping us improve the manuscript.